# The feather degradation mechanisms of a new *Streptomyces* sp. isolate SCUT-3

Zhi-Wei Li [1], Shuang Liang[1], Ye Ke[2], Jun-Jin Deng[1], Ming-Shu Zhang[1], De-Lin Lu[1], Jia-Zhou Li[3] & Xiao-Chun Luo [1 ✉]

Feather waste is the highest protein-containing resource in nature and is poorly reused. Bioconversion is widely accepted as a low-cost and environmentally benign process, but limited by the availability of safe and highly efficient feather degrading bacteria (FDB) for its industrial-scale fermentation. Excessive focuses on keratinase and limited knowledge of other factors have hindered complete understanding of the mechanisms employed by FDB to utilize feathers and feather cycling in the biosphere. *Streptomyces* sp. SCUT-3 can efficiently degrade feather to products with high amino acid content, useful as a nutrition source for animals, plants and microorganisms. Using multiple omics and other techniques, we reveal how SCUT-3 turns on its feather utilization machinery, including its colonization, reducing agent and protease secretion, peptide/amino acid importation and metabolism, oxygen consumption and iron uptake, spore formation and resuscitation, and so on. This study would shed light on the feather utilization mechanisms of FDBs.

[1] School of Biology and Biological Engineering, South China University of Technology, Guangzhou, Guangdong, P. R. China. [2] Yingdong College of Life Sciences, Shaoguan University, Shaoguan, Guangdong, P. R. China. [3] Zhanjiang Ocean Sciences and Technologies Research Co. LTD, Zhanjiang, Guangdong, P. R. China. ✉email: xcluo@scut.edu.cn

Feathers predominantly comprise beta-keratin (approximately 85%), and exists in the currently living birds and their evolutionary ancestors, the feathered dinosaurs. The poultry industry generates increasing amounts of feather waste; according to the Alltech Global Feed Survey, broiler feed production in 2018 was 0.3 billion tons, which estimated to produce about 0.2 billion tons of chicken and more than 10 million tons of chicken feathers. Numerous disulfide bonds make the feather matrix highly durable and resistant to chemical and physical factors[1].

Traditional hydrothermal and chemical treatments currently used in the feather meal industry are energy exhausted and the produced feather meal has low solubility and is difficult to be digested by animals. These high temperature and pressure treatments release large amounts of sulfur and ammonia waste gases, making them unsustainable and polluting. The most promising alternative approaches involve biodegradation using keratinases or bacteria. Numerous feather degrading bacteria (FDB) have been isolated from various genera, including *Bacillus*, *Staphylococcus*, *Enterococcus*, *Streptomyces*, and *Pseudomonas* sp., among others[2,3]. Most FDB need ≥ 5–6 days to completely degrade 1% feather-containing medium and this low efficiency limits their industrial application. A *Pseudomonas otitis* isolate, H11, was recently reported to degrade 1% feather medium with 88.8% efficiency after 48 h, which is the highest efficiency reported to date[4]. Most research on FDB remains at an experimental stage, requiring adjustment before industry-scale applications can be developed.

To understand the feather-degrading mechanisms of FDB, the keratinases they secrete have been studied intensively[2] and bacteria expressing recombinant keratinases constructed. A recombinant KerK strain, *Bacillus amyloliquefaciens* K11, can completely degrade feather plumes in 12 h, but not the feather shaft[5]. While keratinases do have a vital role in feather degradation, the abundant disulfide bonds in feather keratin lead to a tightly packed structure, inaccessible to these enzymes and disulfide bond reduction mechanisms used by FDB are poorly understood. Only metabolically active *Streptomyces pactum* cells have been reported to produce soluble reducing agents; however, the specific reducing agents involved were not identified[6]. Sulfide production is indispensable for dermatophyte nail infection[7] and a free cysteinyl group is essential for feather degradation by *Escherichia coli* expressing recombinant keratinase[8], indicating that similar reducing agents may be produced by FDB.

FDB are opportunistic species and do not need feathers to survive in soil. Their adjustment to feather utilization mode when they encounter feather in the soil is a systematic process involving colonization, secretion of reducing agents and keratinases, matter transportation, and alterations in metabolism/replication/transcription/translation processes. Merely describing keratinases, as in previous reports of FDB, is far from understanding the feather utilization processes of these organisms. Here, a new FDB, *Streptomyces* sp. SCUT-3, was isolated and found to exhibit high-efficiency industrial feather degradation. The product of its feather fermentation can be used as safe fish feed protein source, plant fertilizer, and microorganism culture medium. Ultra-microstructural analysis revealed the colonization and spore dispersion of SCUT-3 on feathers. Further, we sequenced the genome and compared the transcriptomes of SCUT-3 bacteria cultured on LB and feather to identify factors involved in feather degradation and reveal the mechanisms underlying its feather utilization.

## Results
**SCUT-3 efficiently degrades feather to peptides and amino acids**. Using feather powder plates, we identified 27 isolates from soil under a feather pile in Shaoguan (Guangdong, China) able to degrade feather. Among these isolates, *Streptomyces* sp. SCUT-3 was the most efficient; it could completely degrade both the plumage and shafts of white chicken feathers in 36 h in chicken feather medium (CFM) comprising a 1% feather:liquid ratio (Fig. 1b, c), resulting in 93.6 ± 1.5% solid feather matter weight loss (Fig. 1a). White feather matter protein content was determined as 85% using the Kjeldahl method[9], and the total protein content of feather products did not change before and after fermentation in any following experiments. In 1% CFM, of the 85% total dry feather protein, 31.6% was converted to soluble amino acids, 12.7% to soluble peptides, and the remaining 40.7% remained in the undegraded feather residue and bacterial cells (Fig. 1a). Different from other reported FDB, SCUT-3 produced higher levels of soluble amino acids than peptides, indicating that it could secrete efficient terminal peptidases, as well as endopeptidase. These hydrolyzed amino acids and peptides could be used by SCUT-3 to sustain its growth and proliferation.

The feather degradation efficiency of most FDB isolates is tested using 1% CFM, with few reports of their performance in higher feather: liquid ratio media. For industrial application of FDB, low feather: liquid ratios are inefficient, with resulting products requiring further concentration or drying. We found that SCUT-3 could completely degrade feather in 10% CFM in 8 days, with a 50.3 ± 1.4% degradation rate on day 4. Further, in 40% CFM, a 57.3 ± 2.3% degradation rate was observed on day 6 (Fig. 1a) and no feather plumage or shafts were visible in the fermentation broth, which had the appearance of sticky gruel (Fig. 1d, e). The excellent performance of SCUT-3 in high feather: liquid ratio fermentation makes it a competitive reagent for application in the feather biodegradation industry. Further, the high soluble free amino acid and peptide content of its feather fermentation products make them good candidates for use in animal feed, plant amino acid-containing fertilizer, and as a microbial nitrogen source. The amino acid composition, including free amino acids, free amino acids/peptides, and whole proteins in the 40% fermented feather meal produced by SCUT-3, was compared with that of untreated feather (Supplementary Table 1); cystine content reduced from 8.20% to 5.88%, while methionine increased from 0.46% to 1.38%, indicating disruption of disulfide bonds during feather fermentation.

To determine the safety and nutritional value of the feather meal produced by 40% fermentation for use in animal feed, it was added to tilapia feed. After 4 weeks, all fish survived in the group receiving fermented feather meal as a sole protein source (45% addition) and all tissues and organs were normal on anatomical examination, indicating the safety of the fermented feather meal. The feed coefficient of the fermented feather meal group was high (1.90), which is not unexpected, due to its unbalanced amino composition, solely derived from feather protein. The feed coefficient of the fermented feather meal: fish meal (FFM: FM, 22.5%:22.5%) group was 0.85, close to the full fish meal group (0.71) (Fig. 1h), demonstrating that this fermented feather meal is a relatively good protein source for use in animal feed. Additional nutritional tests are needed in future studies to support its rational addition to feed. To determine whether 10% fermented feather broth is good plant fertilizer, it was added to soil-grown (50 mg amino acids of 10% fermented broth in 50 g soil) and hydroponic (50 mg L$^{-1}$ amino acids) rice. Using both methods, the rice grew much more strongly in the group with feather broth added, with more robust leaves (Fig. 1f). The fresh weight of rice cultured in soil with feather broth addition on week 3 was 2.09 times that of controls (Fig. 1i). Interestingly the roots of rice in the fermented feather broth added group were shorter, with many more root hairs, compared with controls (Fig. 1f). Succulent plants also exhibited earlier budding and better growth in broth

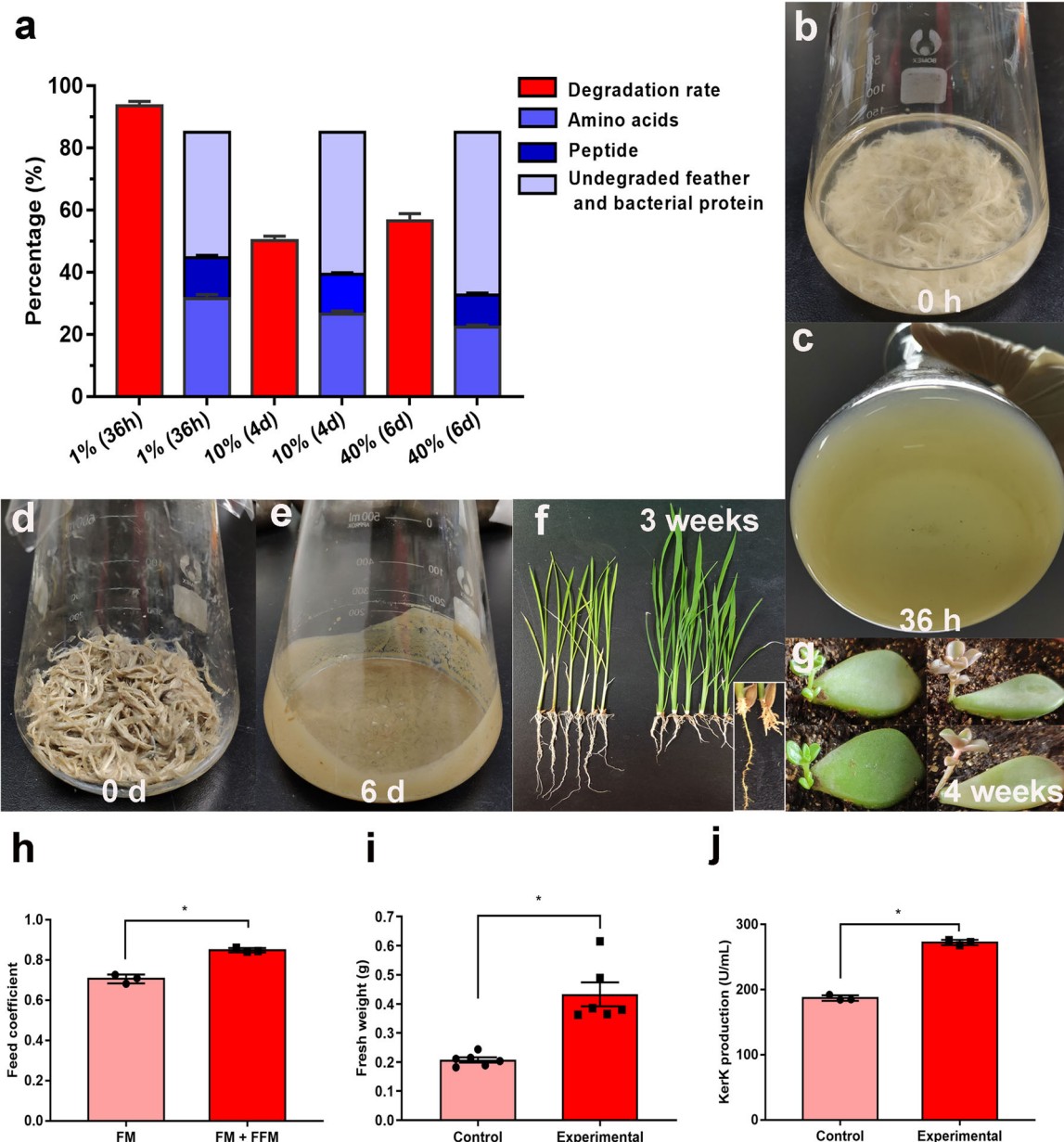

**Fig. 1 High-efficiency degradation of feather by SCUT-3 and application of fermented feather products in animal, plant, and microorganism cultures.**
**a** Feather degradation rate of 1%, 10%, and 40% CFM by SCUT-3, presented as weight loss (red columns), and composition of amino acids, peptides, and insoluble proteins remaining undegraded in feather and bacteria of products (blue columns); $n = 3$/group. **b**, **c** 0 and 36 h SCUT-3 culture in 1% CFM, showing complete feather degradation. **d**, **e** 0 and 6 d SCUT-3 culture in 40% CFM. No feather plumage or shafts were visible in the sticky gruel after 6 days of feather fermentation. **f** Rice growth in control (left) and 50 mg amino acids added (from 10% SCUT-3 cultured CFM broth) (right) soil (50 g), showing the robust leaves and shorter roots with more root hairs. **g** Succulent growth in the control (left) and amino acids added (right) soil; $n = 3$/group, *$P < 0.05$. h Feed coefficients of full fish meal feed (45% fish meal, FM) and half fermented feather meal substitute feed (22.5% fish meal + 22.5% fermented feather meal, FM + FFM) after use for eight weeks in tilapia culture; $n = 20$/group, *$P < 0.05$. **i** Fresh weight of rice grown in control and supplemented soil; $n = 6$/group, *$P < 0.05$. **j** The effect of feather degradation products on microbial recombinant protein production; $n = 3$/group, *$P < 0.05$. The experimental group added 3 mg amino acids (from 10% SCUT-3 cultured CFM broth) to the control LB medium. Asterisks indicate significant differences in feed coefficient, fresh weight and recombinant protein production. $P$-values between groups were obtained by unpaired two-tailed Student's $t$ test. All data were presented as mean ± SD.

supplemented with fermented feather (Fig. 1g). These experiments show that the SCUT-3 fermented feather broth is a good plant amino acid liquid fertilizer. To determine whether the 10% fermented feather broth is benefit for microbial growth and recombinant protein production, it was added to the culture of KerK recombinant *B. subtilis* (3 mg amino acids of 10% fermented broth in 20 mL LB medium). No significant difference

($p = 0.08$) of microbial growth was observed, while the production of recombinant KerK was improved by 45.5% (Fig. 1j); a similar phenomenon was observed in another recombinant esterase *B. subtilis* (Supplementary Fig. 1). These experiments show 10% fermented feather broth can improve microorganism's recombinant protein production, probably by supplying amino acids for protein synthesis.

**SCUT-3 is a new Actinomyces species with penetration ability**. The 16 s rRNA of SCUT-3 was sequenced (Gene Bank accession number MK743936.1) and a phylogenetic tree generated, demonstrating that SCUT-3 is a species of the *Streptomyces* genus, genetically closest to *Streptomyces thermolineatus* (Supplementary Fig. 2). We then assembled the genome of SCUT-3 from Single Molecule Real Time Sequencing data, revealing that its size is approximately 6.08 Mb; further, 5811 genes were annotated (Gene Bank accession number: CP046907). Calculation of average nucleotide identity (ANI) showed that the highest ANI value for SCUT-3 (78.51%) was with *S. cattleya* DSM 46488, among 1222 *Streptomyces* genomes available (Supplementary Table 2), indicating that SCUT-3 is a new species, different from all those tested. The physiological and biochemical characteristics of SCUT-3 were tested and compared with those of *S. thermolineatus* A1484, according to Bergey's Manual of Systematic Bacteriology (Supplementary Table 3). Unlike *S. thermolineatus*, SCUT-3 was positive for hydrogen sulfide production, while mannitol use was negative. SCUT-3 was a Gram-positive bacterium with white colonies on Gauze No. 1 medium plates at day 2, becoming gray-green by day 5. The surface of the colony was dry and had an earthy smell. We compared the feather-degrading capacity of SCUT-3 with those of three other *Streptomyces* genus FDB species: *S. thermolineatus*, *S. fradiae*, and *S. cattleya*; SCUT-3 had the highest feather-degrading ability (Fig. 2g), the highest keratinase activity (Fig. 2h), and second most potent disulfide bond reduction activity (after *S. thermolineatus*) (Fig. 2i). These data suggest that SCUT-3 is a new *Streptomyces* species, which we named *Streptomyces* sp. SCUT-3. The strain has been deposited in the Guangdong Provincial Center for Microbial Strains and the strain number is GDMCC No: 60612.

SEM examination showed that SCUT-3 produced tightly tangled filamentous mycelium around feather barbules on days 1–2 (Fig. 2b). Barbule surfaces were compactly attached by the hyphae, then disintegrated and digested by SCUT-3 on days 2–3 (Fig. 2c). Numerous smooth spores were generated along the filaments at regular intervals on day 3, which then dispersed onto the surface of a different barbule and quickly germinated to form new filamentous mycelia (Fig. 2d–f). Penetration testing showed that SCUT-3 had the strongest mechanical penetration force, followed by *S. thermolineatus*, *S. cattleya*, and *S. fradiae* (Fig. 2j), as it could penetrate most deeply into 4% agar. The efficient elongation and tangling of its mycelium, as well as effective spore generation and diffusion, rapid spore germination, and strong mechanical penetration, may help SCUT-3 to quickly and competitively occupy the feather niche and become the dominant flora after encountering feather in the soil.

**Transcriptomes showed SCUT-3's adaption to feather medium**. Most FDB are opportunistic species that can survive without feather. Hence, it is reasonable to suppose that they live in different modes in the presence or absence of feather; however, no previous report has compared transcriptome differences in detail, according to growth on feather-containing or other media. Here, we found that SCUT-3 only degraded the feather in the feather medium, but not feather added to LB medium, indicating that SCUT-3 does not use feather when other nutrition is available, and that feather utilization-associated machines are shut off in the LB-cultured SCUT-3. Transcriptome of SCUT-3 cultured for 24 h in LB or 1% feather medium were sequenced and compared. The RNA-seq data used in this study are deposited in the National Center for Biotechnology Information SRA database (SRA accession no. PRJNA611875). Then, contigs were assembled and annotated, with reference to the SCUT-3 genome data. We detected 775 up-regulated and 623 down-regulated differentially expressed genes (DEGs) (volcano plot, Supplementary Fig. 3). Significant DEGs ($p < 0.05$) were enriched for specific GO terms (Supplementary Figs. 4 and 5) and KEGG pathways (Supplementary Fig. 6). Next, we mapped the significant DEGs to the SCUT-3 genome, revealing that many were regulated in operon mode (see subsequent results). Sixteen of the top 20 differential pathways were metabolism-related, indicating that SCUT-3 invokes an alternative metabolism strategy when using feather as its sole carbon and nitrogen source, relative to LB medium. The remaining four of the top 20 pathways comprised a two-component system, quorum sensing, oxidative phosphorylation, and ABC transport, reflecting the activities of SCUT-3 in sensing its circumstances, active oxidative respiration, and matter transportation, during feather utilization. Additional DEGs and their organization in the genome are described in detail below, along with further experimental data, revealing the feather utilization mechanisms likely used by SCUT-3 bacteria.

**Disulfide bond reduction is the first key step**. Although few details are known about the mechanisms involved in disulfide bond reduction by FDB, the importance of disulfide bond sulfitolysis for feather degradation is established. Based on release of sulfhydryl content from oxidized glutathione, we found that only living SCUT-3 cells could reduce disulfide bonds, while its extracellular secretory supernatant or intracellular cell lysate could not (Fig. 3b), similar to previously reported FDB[6].

Production of sulfite to reduce disulfide bonds is important for dermatophyte infection of nails[7]. Here, we found that 80 mM $Na_2SO_3$ addition could efficiently improve feather degradation of the recombinant keratinase KerK in vitro (from $11.5 \pm 0.7\%$ to $15.5 \pm 1.1\%$, $p = 0.006$) (Fig. 3c), the degradation rate of 8 M $Na_2SO_3$ addition could reached much higher of $43.5 \pm 0.7\%$, while there was no effect using <80 mM $Na_2SO_3$ (Supplementary Fig. 7). Sulfite production in feather cultured SCUT-3 supernatant was confirmed by $BaCl_2$ precipitation, HCl bubble production, and $KMnO_4$ bleaching (Fig. 3a), suggesting that this process likely accompanies disulfide bond destruction by SCUT-3. Although the concentration in the culture supernatant may not reach as high as 80 mM, we believe comparatively high sulfite concentrations could be produced locally in the compact contact surfaces between SCUT-3 mycelia and feather barbules detected by SEM (Fig. 2b–c). Dermatophyte mutants of *cdo* and *ssu* (genes encoding proteins involved in sulfite production and exportation) lose their nail infection ability[7]. Two cysteine dioxygenase genes, *cdo1* and *cdo2*, were up-regulated in feather medium-cultured SCUT-3 according to transcriptome data and confirmed by real-time RT-PCR, with fold-change values, 50.2 and 8.3, respectively. An L-methionine γ-lyase, *mdeA*, was also up-regulated (28.6-fold) (Fig. 3d). Cdo and MdeA can oxidize the sulfhydryl group of cysteine and the methyl thioether of methionine, respectively, to generate sulfite[10,11]. No *ssu* gene was detected in the SCUT-3 genome; however, another sulfite exporter, *tauE*, was found and up-regulated approximately 2.0-fold (Fig. 3d). These data indicate that sulfite production may be involved in SCUT-3 breakdown of feather disulfide bonds.

Besides sulfite, free cysteinyl groups are important reducing agents in feather sulfitolysis. Cysteinyl-glycine produced from GSH is required for feather degradation by *E. coli* expressing recombinant keratinase[8]. DTNB assays revealed the participation of free cysteinyl in SCUT-3-mediated sulfitolysis, with addition of DTNB (5 mM) reducing the SCUT-3 feather degradation rate (Fig. 3c). Many actinomycetes lack GSH, with mycothiol (AcCys-GlcN-Ins, MSH) functioning as a surrogate[12]. Two genes (*mshA* and *mshD*) involved in mycothiol synthesis were up-regulated 4.3- and 9.8-fold, respectively, in feather medium-cultured

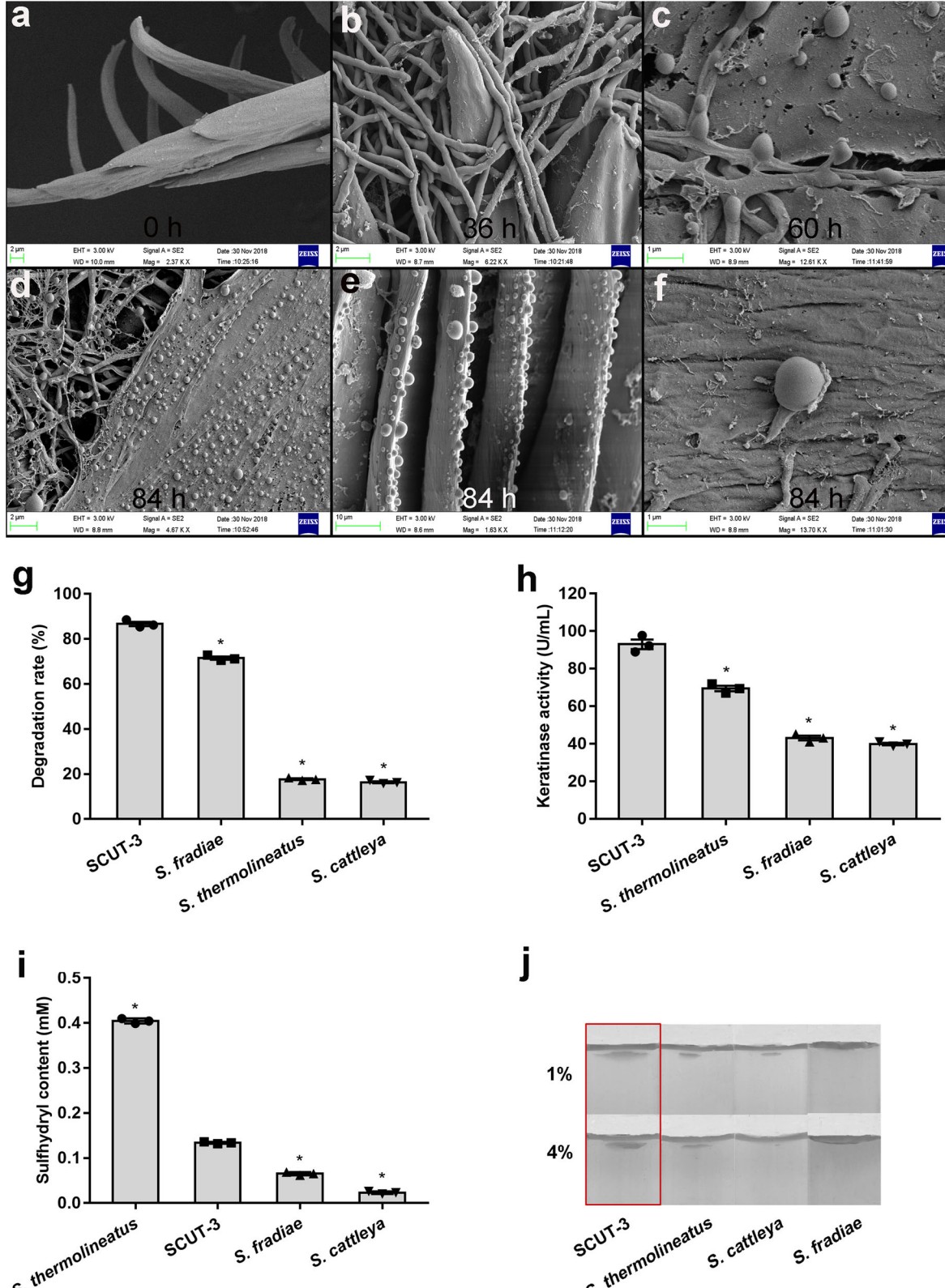

**Fig. 2 Scanning electron microscopy observation of SCUT-3 cultured on feather and comparison of feather degradation by SCUT-3 and three other Streptomyces sp. FDB. a** Untreated chicken feather control. **b** Feather cultured with SCUT-3 for 36 h, showing the mycelium tangled around barbules. **c** Feather cultured with SCUT-3 for 60 h, showing the compact attachment of the mycelium and the digestion and slit of the barbule surface by SCUT-3. **d**, **e** Feather cultured with SCUT-3 for 84 h, showing the formation, diffusion of SCUT-3 spores. **f** Spores diffusion is accompanied by spore resuscitation, beginning a new growth cycle. **g–i** Comparison of the degradation rate, keratinase activity, reducing power; $n = 3$/group, *$P < 0.05$. **j** Penetration of SCUT-3 and three other Streptomyces sp. FDB in 1% and 4% LB agar. $P$-values between groups were obtained by unpaired two-tailed Student's $t$ test. All data were presented as mean ± SD.

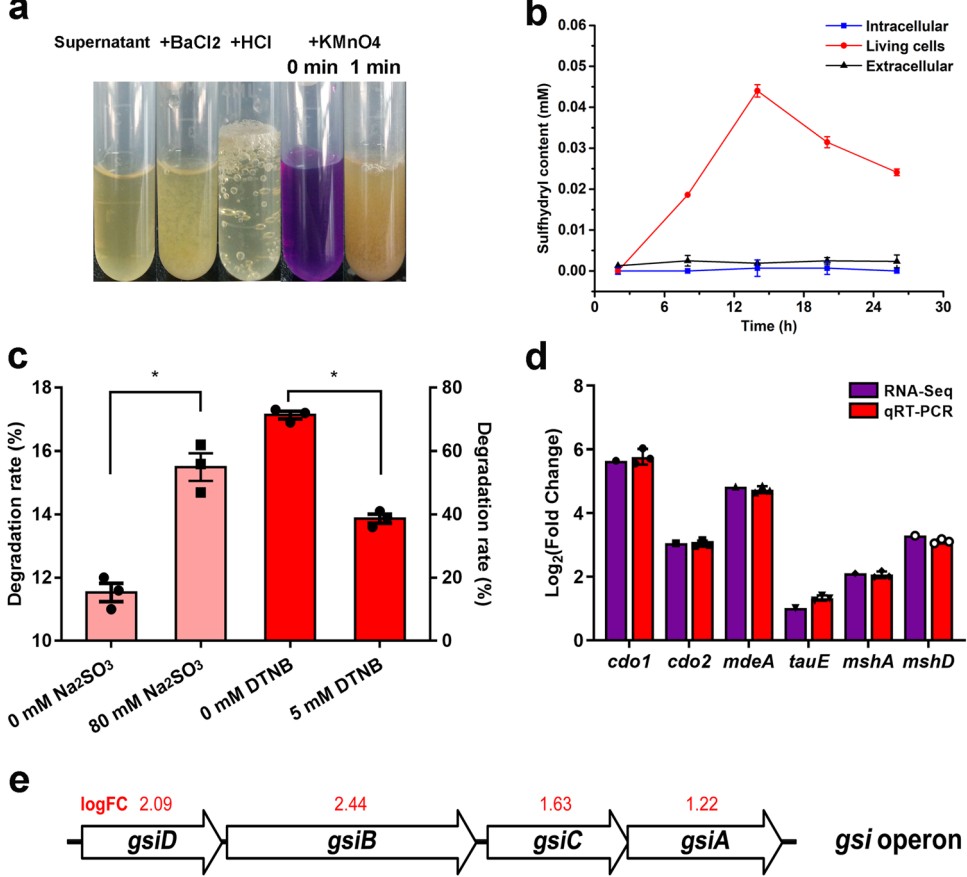

**Fig. 3 Feather disulfide bond reduction and up-regulation of reducing agent production-associated genes in feather medium-cultured SCUT-3. a** Sulfite detection in SCUT-3-cultured CFM supernatant by $BaCl_2$ precipitation, HCl bubble production, and $KMnO_4$ bleaching tests. **b** Evaluation of SCUT-3 reducing power by detection of sulfhydryl content release from oxidized GSH, showing that only the living cells, but not its intracellular or extracellular components, exhibited reducing power; $n = 3$/group. **c** Sulfite improvement of KerK (pink) and DTNB inhibition of SCUT-3 (red) feather degradation, showing the function of sulfite and sulfhydryl groups in feather disulfide bond breakdown; $n = 3$/group, *$P < 0.05$, $P$-values between groups were obtained by unpaired two-tailed Student's $t$-test. d qRT-PCR verification of the up-regulation of sulfite production (*cdo1/cdo2/mdeA*), sulfite exportation (*tauE*), and mycothiol synthesis (*mshA/mshD*) genes ($n = 3$/group). e The operon structure of glutathione transporter genes (red numbers above gene symbols are $log_2$ FC values for each gene). All data were presented as mean ± SD.

SCUT-3 (Fig. 3d). No MSH transporter is yet defined[13]. A GSH transporter operon, *gsiDBCA*, was annotated in the SCUT-3 genome and its expression up-regulated in feather medium-cultured SCUT-3 (Fig. 3e). Whether MSH is the cysteinyl group involved in feather sulfitolysis by SCUT-3 and whether *gsiDBCA* is responsible for free cysteinyl group transportation in SCUT-3 requires further study.

**Protease hydrolyzation and peptide/amino acid transportation.**
Following disulfide bond reduction, feather is further hydrolyzed by secretory proteases, to generate peptides and free amino acids. SCUT-3 secreted abundant proteases and the keratinase activity of 24 h 1% feather medium-cultured SCUT-3 supernatant was 65.8 U mL$^{-1}$, about 4.6 times that of LB-cultured SCUT-3 (14.3 U mL$^{-1}$) (Table 1). In transcriptome data, 19 of 22 significantly ($p < 0.05$) differentially expressed extracellular protease genes were up-regulated in feather medium-cultured SCUT-3, with fold-change values much higher than the remaining three down-regulated proteases (Fig. 4a). The 19 up-regulated proteases were serine-type, cysteine-type, and metalloproteases; most were endopeptidases, and one dipeptidyl-peptidase and one carboxypeptidase were identified. These proteases' classifications according to MEROPS database and Pfam peptidase domains were shown in Supplementary Table 4. Unlike changes to the

expression pattern of extracellular proteases, 10 up-regulated and 11 down-regulated intracellular proteases were detected. Expression levels of the six most up-regulated proteases were verified by qRT-PCR, with the highest fold-change in expression for Sep39 protease, of 451.9-fold (Fig. 4b). These extracellular proteases are likely involved in keratin hydrolysis to produce peptides and amino acids by SCUT-3.

To elucidate the expression pattern of these proteinases during the feather degradation, qPCR of the nine most up-regulated proteases were detected at 0, 3, 6, 12, and 24 h in 1% CFM medium culture. As we showed in the heatmap of Fig. 4c, all nine tested proteases were quickly and up-regulated at 3 h. Among them, the up-regulation fold of *sep39* kept rising until 24 h, that of *cp17* continued to rise until 12 h and went down at 24 h. The up-regulation folds of the other seven proteases sustained at their 3 h levels and went down at the later different time points. The different up-regulation mode of these proteases reflected their cooperation mode during the feather degradation. Besides proteases, we also detected the *cdo1*'s expression (Fig. 4c) and found up-regulation folds of Cdo1 kept rising during 24 h as Sep39, which indicates the disulfide reduction accompanies the peptide bond hydrolysis in the feather degradation process.

To further confirm the function of the up-regulated proteases in the feather degradation, we overexpressed them in SCUT-3. In

**Table 1 Overexpression of proteinase Sep39 in *Streptomyces* sp. SCUT-3.**

| Medium | Strain | Relative mRNA level | Keratinase activity (U mL$^{-1}$) | Degradation rate (%) | Peptide content (mg mL$^{-1}$) | Amino acid content (mg mL$^{-1}$) |
|---|---|---|---|---|---|---|
| LB(24 h) | SCUT-3 | 1 | 14.3 ± 1.9 | - | - | - |
| | SCUT-3-sep39 | 5.6 ± 0.1* | 64.4 ± 2.1* | - | - | - |
| 1%CFM (24 h) | SCUT-3 | 261.1 ± 41.3 | 65.8 ± 5.0 | 64.5 ± 1.1 | 1.3 ± 0.1 | 1.4 ± 0.1 |
| | SCUT-3-sep39 | 618.9 ± 91.5* | 102.5 ± 3.8* | 71.0 ± 1.0* | 1.5 ± 0.1* | 1.7 ± 0.1* |
| 5%CFM (48 h) | SCUT-3 | - | 69.1 ± 3.7 | 45.5 ± 1.1 | 4.5 ± 0.2 | 6.5 ± 0.2 |
| | SCUT-3-sep39 | - | 117.1 ± 3.1* | 53.3 ± 0.8* | 5.8 ± 0.6* | 8.7 ± 0.6* |

Note: Asterisks indicate significant differences compared to wild-type SCUT-3; $n = 3$/group, *$P < 0.05$. P-values between groups were obtained by unpaired two-tailed Student's $t$ test. All data were presented as mean ± SD.

this article, we showed the overexpression results of the highest-up-regulated protease Sep39. As we showed in Table 1, the 24 h *sep39* mRNA level of the overexpression strain SCUT-3-*sep39* cultured in LB medium was 5.6 times of wild-type SCUT-3 and its corresponding keratinase activity was about 4 times as wild-type bacteria, which indicates *sep39* had been successfully overexpressed in SCUT-3-*sep39*. Applied this reconstructed SCUT-3-*sep39* in 1% and 5% CFM's feather degradation, we found that not only the *sep39* mRNA level and keratinase activity were up-regulated compared to wild-type strain in both trials, but also the degradation rate and peptide/amino acid content in the degraded feather medium supernatant were improved in SCUT-3-*sep39* group (Table 1). These data confirmed Sep39 is an important keratinase for SCUT-3's feather degradation and its overexpression can improve SCUT-3's feather degradation efficiency. Other proteases' overexpression strains are now under construction in our lab and the pilot experiments showed their overexpression could make different degree's improvement of the SCUT-3's feather degradation efficiency. Another exciting phenomenon is that co-overexpression of Sep39 and Cdo1 in SCUT-3 achieved a higher degradation efficiency than single overexpression of Sep39 itself, which also indicated the cooperation of disulfide bond reduction and peptide bond hydrolysis during feather degradation. These data will be published in our next article.

Hydrolyzed peptides and amino acids can be imported by bacteria as nutrients via two categories of transporters: the proton motive force-driven transporters (PTR transporters) and ATP binding cassette-containing transporters (ABC transporters)[14]. In this study, two PTR di- and tri-peptide transporters, *cstA* and *dtpT*, were identified as up-regulated in feather medium-cultured SCUT-3 (Fig. 4d). Further, two peptide ABC transporter operons, *oppABCDF* I and *oppABCDF* II, were up-regulated (Fig. 4d). Two amino acid ABC transporter operons, *metNIQ* and *livJHMGF*, responsible for Met and branched chain amino acid (Ile, Leu and Val) importation, were also up-regulated (Fig. 4d). Up-regulation of these transporter genes and operons indicates efficient peptide and amino acid importation by SCUT-3 to maintain its growth on feather.

**Iron uptake are regulated by the Fur regulon.** Iron is essential element for organisms, participating oxygen transport, ATP generation, cell growth and proliferation, ribonucleotide reduction, and so on. Ferric uptake regulator (Fur) is an important regulator for bacterial iron uptake. Holo-Fur with Fe$^{2+}$ binds its target DNA operators and functions as a repressor. Under iron limited condition, apo-Fur dissociates from DNA and the repression is relieved. In this study numerous genes encoding iron-utilizing proteins in the Fur regulon were up-regulated in white CFM-cultured SCUT-3, including the siderophore staphyloferrin B synthetase *SbnA*, siderophore bacillibactin synthesis operon *DhbACEBF*, siderophore enterobactin secretion exporter *entS*, catecholate siderophore detoxification gene *catE*, elemental iron transporter operon *EfeUOB*, ABC transporter *yfiYZ*/*yusV* ATPase[15–20] (Fig. 5a, c), indicating that there was insufficient iron for SCUT-3 growth in white feather medium. To determine whether iron uptake could improve SCUT-3 feather degradation, we added 1 µM FeSO$_4$ to white chicken or duck feather, and brown chicken feather-containing media. Interestingly, iron addition improved the rate of white chicken (77.0%, $p = 0.00003$) and white duck (50.8%, $p = 0.00005$), but not brown chicken ($p = 0.5$), feather degradation by SCUT-3 (Fig. 5d), as the brown color of the feather is caused by iron. This finding is important because white feather broilers are the most cultivated globally.

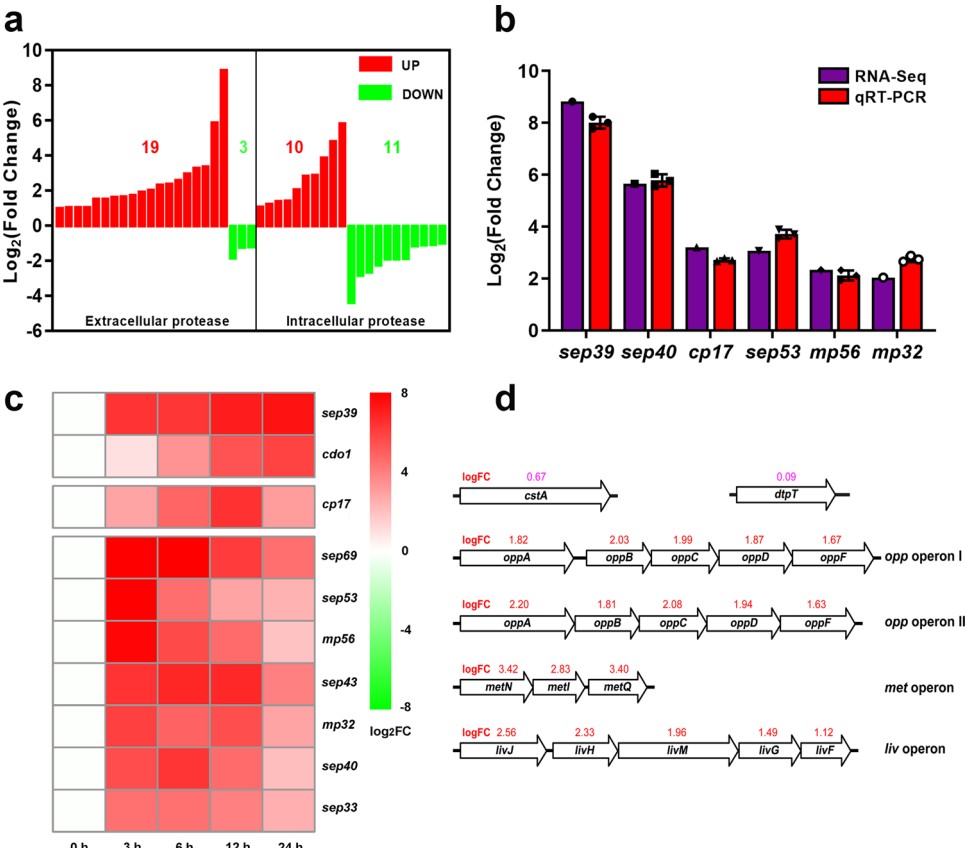

**Fig. 4 Up-regulation of protease secretion and amino acid/peptide transporter genes in feather medium-cultured SCUT-3. a** Significantly regulated extracellular and intracellular protease genes identified by transcriptome analysis, showing the significant up-regulation of most secretory proteases. **b** qRT-PCR verification of the top 6 significantly up-regulated protease genes; $n = 3$/group. **c** qPCR results of nine protease genes and one disulfide reduction related gene at 0, 3, 6, 12, and 24 h; $n = 3$/group. **d** The up-regulation of two PTR di- and tri-peptide transporter genes (*cstA* and *dtpT*), two peptide ABC transporter operons (*oppABCDF* I and *oppABCDF* II), and two amino acid ABC transporter operons (*metNIQ* and *livJHMGF*). Numbers above genes are $\log_2$ FC values for each gene; red > 1.0, pink < 1.0. *P*-values between groups were obtained by unpaired two-tailed Student's *t* test. All data were presented as mean ± SD.

**Aeration improves SCUT-3 feather degradation efficiency.** Most *Streptomyces* are aerobic bacteria. A series of genes and operons involved in oxygen sensing and the respiratory chain were up-regulated in feather medium-cultured SCUT-3 (Fig. 5b, e). The oxygen sensor, *AcrB*[21], and the hypoxia/oxygen sensor, *dosT*[22], were up-regulated. In addition, the following electron transport chain-associated genes were up-regulated: the 14-gene NADH dehydrogenase 1 operon, *NuoA-N* (complex I); the NADH dehydrogenase 2 gene, *ndh2*; cytochrome c oxidase, *ctaD*; and the ATP synthase, *atpIBEFHAGDC* (complex V)[23,24]. Besides these genes, enhanced iron uptake (as described above) may also provide ferric ion for the Fe-S clusters of those proteins involved in the electron transport chain. An aerating experiment showed that aeration could improve SCUT-3 feather degradation efficiency by 27.5% in 1% feather medium and 99.3% in 10% feather medium (Fig. 5f), consistent with up-regulation of electron transport chain genes, demonstrating that SCUT-3 feather degradation is an oxygen-consuming process.

**Active metabolism, spore formation/resuscitation, and division.** In addition to disulfide bond reduction, protease hydrolyzation, peptide and amino acid uptake, iron uptake, and oxygen consumption by SCUT-3 during feather degradation, more interesting details regarding how SCUT-3 uses feather material for its growth and cell division were deduced from comparative analysis of transcriptome data. We determined that the following metabolic processes were active in SCUT-3 grown in feather medium. Uptake and synthesis genes for vitamins involved in catabolism were up-regulated, including the thiamine (vitamin B1, for dehydrogenase) transporter, *ykoEDC*; the thiamine synthesis genes, *thiC/thiG/thiS*; the riboflavin (vitamin B2, for redox reaction) synthesis operon, *ribDEBAH*; the pyridoxal phosphate (vitamin B6, for transamination) synthesis genes, *pdxH1/pdxH2*; the nicotinamide riboside (vitamin B3, for redox reaction) transporter, *PnuC*; the NAD biosynthesis gene, *NadR*[25]; the pantothenate (vitamin B5, precursor for CoA) synthesis genes, *panB/panC/panD*; and the CoA synthesis gene, *coaBC* (Supplementary Fig. 8a). Genes involved in amino acid catabolism pathways were also up-regulated, including the arginine: pyruvate transaminase, *aruH*; the aromatic amino acid metabolism genes, *paaK/paaABCDE/paaI/paaJ*; and genes involved in branched chain amino acid metabolism, *bkdABC* (Supplementary Fig. 8b).

In addition to its 85% keratin content, feather also comprises approximately 5% lipids. Genes involved in lipid digestion and catabolism were also up-regulated on feather medium, including extracellular esterase (*estB*), cell-wall associated carboxylesterase (*caeB*), the glycerol uptake operon (*glpFKD*), the long chain fatty acid-CoA ligases (*lcfB4/lcfB8/fadD15*), fatty acid β-oxidation (*fadA/fadE10*), Acetyl CoA synthetase (*acsA3/acsA4/acsA5*), and steroid and cholesterol catabolism (k*sdD/hsaA1/hsaA2-pchF-ntaB/hsaA3-hsaC/hsaB*)[26] (Supplementary Fig. 8c). Based on

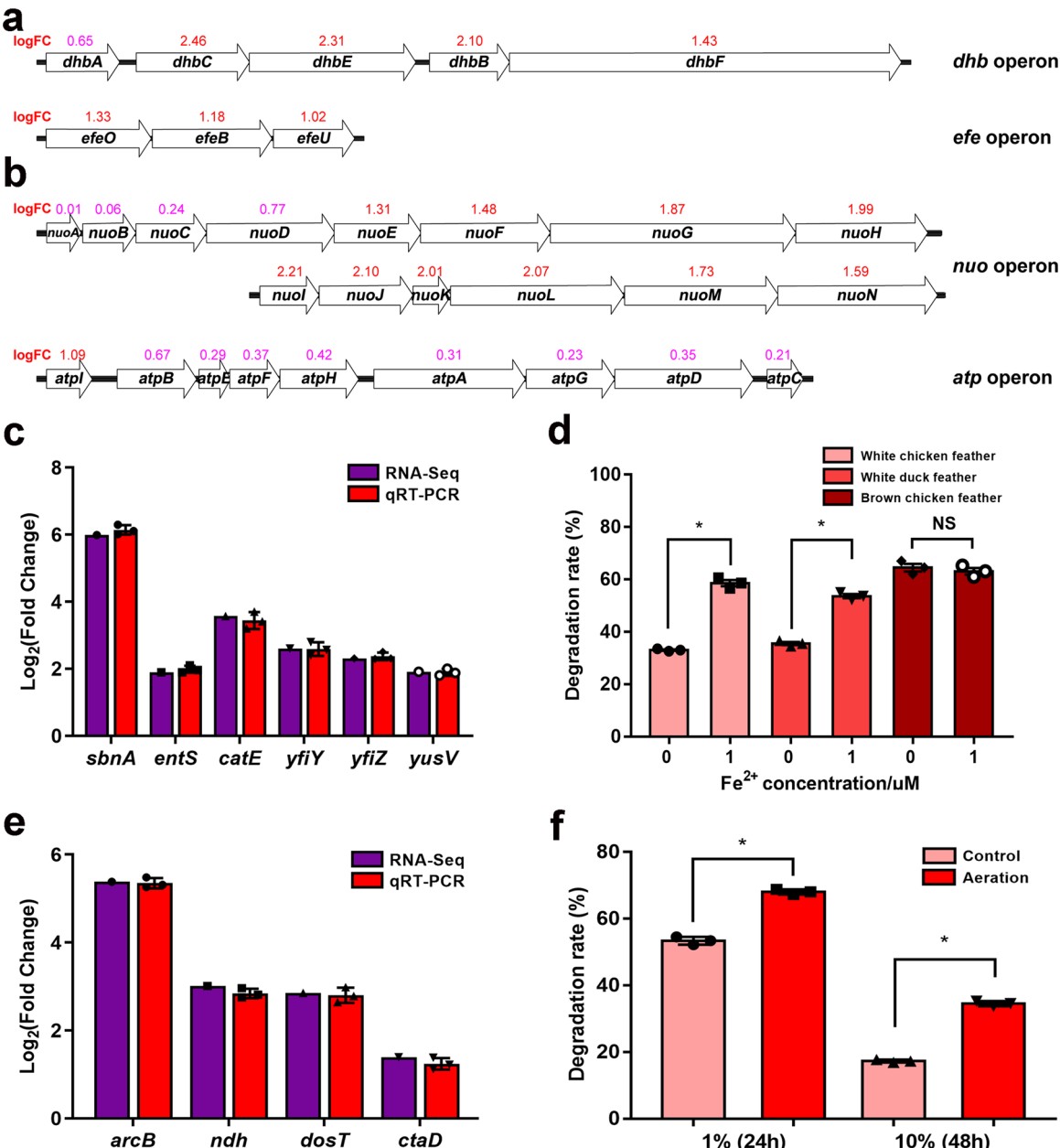

**Fig. 5 Iron uptake and oxygen consumption of feather medium-cultured SCUT-3. a, b** Up-regulation of the siderophore synthesis (*DhbACEBF*), iron transporter (*EfeUOB*), NADH dehydrogenase 1 (*NuoA-N*), and ATP synthase (*atpIBEFHAGDC*) operons; numbers above genes are the $\log_2$ FC values for each gene (red > 1.0, pink < 1.0). **c** qRT-PCR verification of siderophore synthesis and exporter (*SbnA, SbnA, catE*) and iron transporter (*yfiY, yfiZ, yusV*) gene expression; $n = 3$/group. **d** FeSO$_4$ addition (1 μM) improved degradation of white chicken and duck feathers by SCUT-3, but not brown chicken feathers (NS means no significant, $P = 0.52$); $n = 3$/group, *$P < 0.05$. **e** qRT-PCR verification of oxygen sensor (*AcrB/dosT*), NADH dehydrogenase 2 (*ndh2*), and cytochrome c oxidase (*ctaD*) gene expression; $n = 3$/group. **f** Aeration improved SCUT-3 feather degradation in 1% and 10% CFM; $n = 3$/group, *$P < 0.05$. $P$-values between groups were obtained by unpaired two-tailed Student's $t$ test. All data were presented as mean ± SD.

these data, alongside the active electron transport and oxidative phosphorylation, we conclude that SCUT-3 can efficiently catabolize both amino acids and lipids to provide energy and metabolites for its growth in feather medium.

Numerous genes involved in DNA replication and cell division were also up-regulated on feather medium, including the purine uptake permease, *pbuG*; genes required for de novo purine synthesis, *purS-purQ-purL/guaB1/guaB4/guaD*; the pyrimidine de novo synthesis operon, *pyrB -pyrC-carA-carB-pyrD-pyrF*; DNA helicases, *helD/dnaB*; DNA polymerases, *dnaX/dnaQ*; dimer chromosome segregation genes, *xerC/xerD*; and the cell division regulator, *yofA* (Supplementary Fig. 8d). Aerial mycelium,

septum, and spore formation associated genes (*afsR1/afsR3/afsR8/afsK/ramA/whiB1/whiB7*)[27–29] were also up-regulated. Three spore resuscitation-promoting factor (*rpf1/rpf2/rpfA*) implicated in the cleavage of dormant cell walls and subsequent promotion of growth and metabolic reactivation[30] were up-regulated. Combined with observation of SCUT-3 morphology by SEM, it is clear that this bacterium can grow efficiently on feather by prompt mycelium formation, efficient DNA replication and spore formation, and rapid spore diffusion and resuscitation. Rapid DNA replication can also lead to intensive DNA errors. LexA is a transcriptional repressor that inhibits SOS response genes. We found that *lexA* was down-regulated and that the DNA

damage repair associated gene, *dinF*, the recombination repair genes, *recA/recX/recD*, the uvrABC system genes, *uvrA1/uvrA2/uvrA4*, the mismatch repair gene, *mutL*, and the non-homologous end-joining repair gene, *ligA*, were up-regulated to ensure genetic fidelity during rapid cell division (Supplementary Fig. 8d).

In addition to DNA replication genes, factors associated with transcription and translation were also up-regulated on feather medium, including the transcription termination gene, *rho*; various amino acid tRNA genes, tRNA-Ala/Arg/Met/Glu/Gln/ Leu/Thr/Pro; tRNA pseudouridine synthase, *truB*; tRNA processing, *rbn*[31]; tRNA repair, *rtcB*[32]; amino acid tRNA ligase, *leuS/ hisS*; misacylated tRNA proofreading, *ybaK*[33]; ribosomal proteins, *rpsO* (S15), *rpmF* (L32), *rpmB* (L28), *rpmH* (L34), and *rplM* (L13); ribosome maturation, *rbfA/rsgA/rimM* (30 S), *rlmCD* (23 S rRNA); ribosomal protein acetylation, *ydaF2/ydaF3/ydaF7/ydaF8*; ribosomal assembly and disassembly, *hflX/rhlE3/rhlE4*[34,35]; translation initiation, *infB*; stalled ribosome rescue, *arfB*[36]; and translation termination and protein release, *prmC*[37] (Supplementary Fig. 8e). Up-regulation of these genes indicates the intensive protein synthesis required for SCUT-3 growth on feather.

## Discussion

The traditional physical/chemical treatment methods for feather waste are gradually being abandoned because of the resulting pollution and amino acid destruction. Keratinases are considered an alternative green approach and have attracted intensive

research[38]. Our group also spent an inordinate amount of time developing a high-efficiency recombinant keratinase, which was ultimately not a fruitful endeavor. We produced high levels of the keratinase, KerK (approximately 1000 U mL$^{-1}$) as did another group[5]; however, this approach was ultimately disappointing, since without a sulfitolysis agent, keratinase feather hydrolyzation has very low efficiency and the cost of recombinant keratinase preparation is considerable. Microbial fermentation is the most economical method and there have been many attempts to isolate natural FDB; however, most FDB is insufficiently efficient for application in industry-scale feather hydrolysis[1,39,40]. Keratinase overexpressing bacteria have been generated that can hydrolyze feather using their own reducing power, with limited success. The KerK overexpressing *B. subtilis* constructed by our group hydrolyzes feather much more efficiently than wild-type *B. subtilis*; however, its efficiency remains unsatisfactory.

Microbial feather utilization is a systematic process, involving disulfide bond reduction and keratinase hydrolyzation, bacterial colonization, import of hydrolyzed peptides and amino acids, and metabolism and growth on feather material. Many of these processes have been overlooked with the excessive focus on keratinases.

Based on our findings, we present a schematic illustrating the possible feather utilization mechanisms of the new isolate, *Streptomyces* sp. SCUT-3 (Fig. 6), following the logic outlined below. SCUT-3 does not degrade feather in nutritionally rich LB medium; therefore, we speculate that SCUT-3 initiates activation of its feather degradation machinery in the comparatively limited

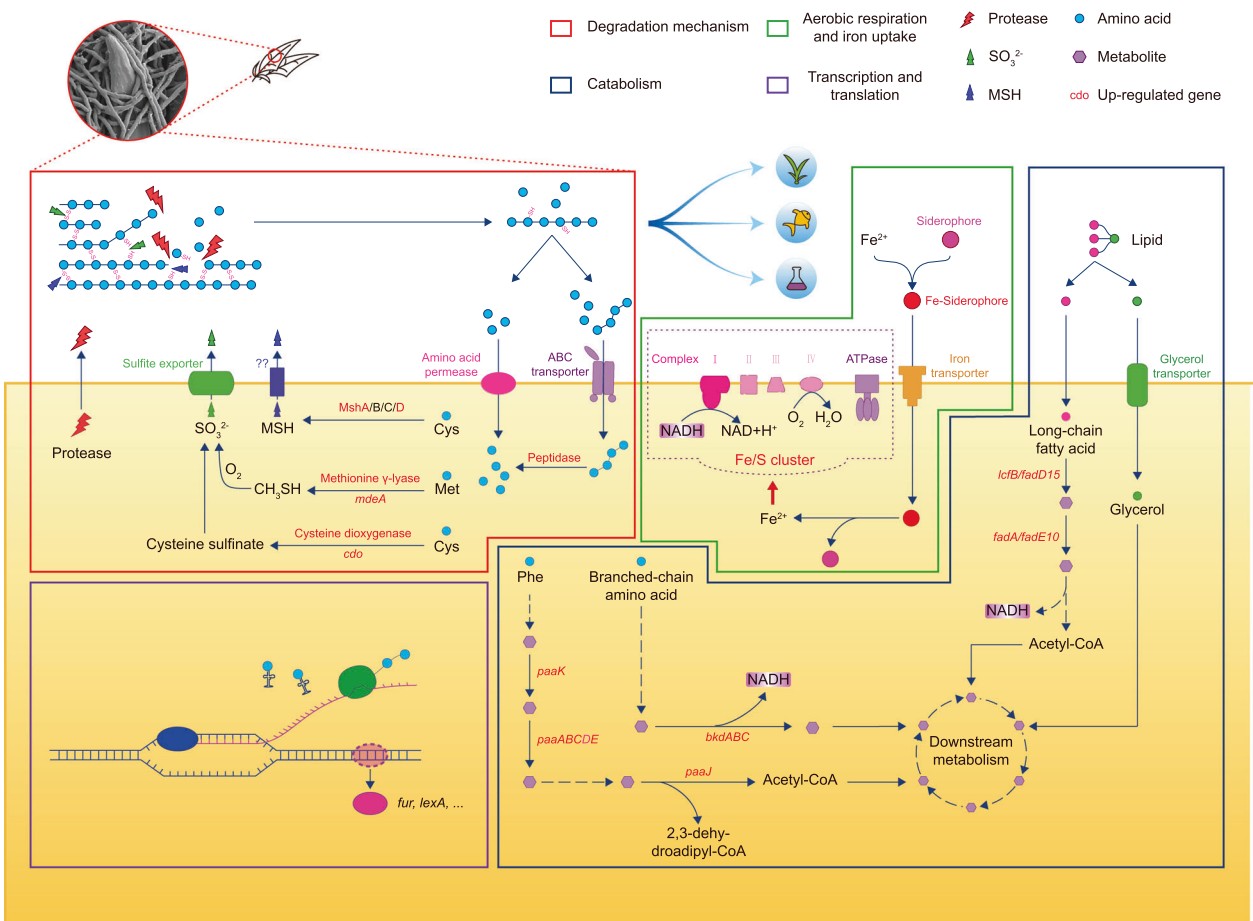

**Fig. 6 Schematic diagram of SCUT-3 feather utilization mechanisms.** Reactions framed in red are processes potentially involved in feather disulfide bond breakdown, peptide bond hydrolysis, and amino acid importation. The hydrolyzed feather products could be used as a nutrition source for plants, animals, and microorganisms (blue arrows). Reactions framed in blue include catabolism of amino acids and lipids from feather to produced metabolites and NADH. Reactions framed in green include oxidation of NADH and ATP synthesis, while those in the purple frame are involved in gene transcription regulation.

nutrition in soil when it encounters a molted bird feather. Sulfite and free cysteinyl groups are secreted to reduce the disulfide bonds in the feather keratin and proteases released to access the peptide bonds of keratin, hydrolyzing them to generate peptides and amino acids, which are imported into the cell via the up-regulated peptide and amino acid transporters, where peptides can be further hydrolyzed by intracellular proteases. Feather lipids are also hydrolyzed by secreted or cell-wall bound esterase, while glycerol, fatty acids, and other lipids (such as cholesterol) are imported by their associated transporters. Some amino acids and lipids are catabolized to generate different metabolites for anabolism and NADH for electron transportation and oxidative phosphorylation, to yield energy for further cell division and growth. Purine and pyrimidine are synthesized for DNA replication and RNA (rRNA, tRNA, and mRNA) synthesis. Spores are formed and dispersed to new feather barbules and resuscitated to undergo further cycles of feather degradation. Many details in our schematic require verification. The inclusion of genes that may be involved in these processes was prompted by their up-regulation; hence they are candidate targets for genetic manipulation of this isolate, to further enhance its feather degradation efficiency. The model we present provides a reference for mechanisms of feather degradation by other soil-borne FDB, and will assist under-standing of the cycle of the feather in the biosphere.

Most importantly, this study presents a promising bacterium for green industrial processing of both chicken and duck feather waste. Using 10% submerged and 40% solid-stage fermentation, we achieved high degradation rates of 50.3 ± 1.4% and 57.3 ± 2.3% in 4 days and 6 days, respectively (Fig. 1a), which is unprecedented in previous reports. Compared with the high price of fish meal (1,200 USD ton$^{-1}$) and soybean meal (500 USD ton$^{-1}$) in China, the price of feathers is <100 USD/ton. The estimated cost of feather meal produced by 40% solid fermentation is <200 USD ton$^{-1}$. Further, the meal generated contains a high soluble amino acid (22.4%) and peptide (10.3%) content, and could efficiently replace fish meal in fish feed, as we demonstrated using cultured tilapia. No bio-degraded feather meal has previously been reported to be used in animal feed. Relative to the high price (about 700 USD ton$^{-1}$) and low feed addition effects of feather meal produced using traditional physical-chemical treatment methods, the feather meal produced by our approach is much more profitable. The use of microbe-inoculated feather compost as plant fertilizer was recently reported for growing cherry tomato, resulting in higher fruit yield and better taste, while the duration of composting was very long (90 days)[41]. Here we produced 10% submerged feather fermentation broth in 4 days, which could double the fresh weight of rice plants, indicating great potential as plant amino acid liquid fertilizer. Moreover, 10% submerged feather fermentation broth was also demonstrated to improve the recombinant protein production of microorganisms.

Two key factors, iron addition and aeration, improved the efficiency of SCUT-3 feather degradation. Additional optimization of fermentation, such as agitation and improved methods for supplying air to the fermenter, can likely increase degradation efficiency further. Ongoing experiments in our laboratory include modification of potential SCUT-3 target genes identified in this study, including those involved in sulfite exportation and proteases, among others. These genetic manipulations may also improve the feather degradation efficiency of SCUT-3.

## Methods

**Strains, media, materials, and reagents**. *Streptomyces* sp. SCUT-3, isolated from soil containing disposed feathers by our laboratory was used in the present research. *Streptomyces fradiae* ATCC 10745, *Streptomyces thermolineatus* ATCC 51534, and *Streptomyces cattleya* ATCC 35852 were obtained from China General Microbiological Culture Collection Center (CGMCC) and Guangdong Microbial

Culture Collection Center (GDMCC), and maintained under their respective designated culture conditions. SCUT-3 was cultured in Gauze No. 1 Medium (containing, per liter distilled water: 20.0 g soluble starch, 0.5 g sodium chloride, 0.01 g ferrous sulfate, 1.0 g potassium nitrate, 0.5 g dipotassium hydrogen phosphate, 0.5 g magnesium sulfate, 15.0 g agar; pH 7.2). *Bacillus subtilis* WB600-*kerK*, a strain expressing recombinant keratinase constructed and stored in our laboratory. *Bacillus subtilis* L25, a strain expressing recombinant esterase constructed and stored in our laboratory. Overexpression plasmid pSET152 and conjugation strain *E. coli* ET12567/PUZ8002 were deposited in our laboratory.

Chicken feathers (white and brown) and duck feathers (white) were obtained from the local market, rinsed in double distilled water until completely clean, dried and stored for further study[42]. FDB enrichment media contained (g L$^{-1}$) NH$_4$Cl 0.5 g, NaCl 0.5 g, K$_2$HPO$_4$ 0.3 g, KH$_2$PO$_4$ 0.4 g, MgCl$_2$ 0.1 g, yeast extract 1.0 g, chicken feather 10.0 g; pH 7.5. Feather powder plates contained (g L$^{-1}$) K$_2$HPO$_4$ 1.5 g, MgSO$_4$•7H$_2$O 0.025 g, CaCl$_2$ 0.025 g, FeSO$_4$ 0.015 g, chicken feather powder 10.0 g, agar powder 20.0 g; pH 7.5. Basal medium (BM) contained (g L$^{-1}$) NaCl 0.5 g, KH$_2$PO$_4$ 0.4 g, K$_2$HPO$_4$ 0.3 g; pH 7.2–7.5. CFM was obtained by adding different amounts of chicken feathers to BM (g/100 mL).

Rice (*Oryza sativa*) seeds, succulent (*Sedum Alice Evans*) and cultivated soil were purchased from the local market and farm. Nile Tilapia (*Oreochromis niloticus*) was purchased from Guangdong Tilapia Fine Germchit Field (Guangdong, China). Tilapia nutrition experiment was performed in the Aquaculture Laboratory of College of Marine Science, South China Agricultural University, and was approved by the Experimental Animal Ethics Committee of South China Agricultural University.

Keratin was obtained from J&K Chemical Co., Ltd. (Shanghai, China). Folin phenol reagent, dithiothreitol (DTT), oxidized glutathione (GSSG), 5, 5′-dithiobis (2-nitrobenzoic acid) (DTNB), ninhydrin, and trichloroacetic acid (TCA) were from Sigma (Shanghai, China). Unless specified, all other substrates, chemicals, and primers were purchased from Sangon Biotech (Shanghai, China).

**Screening of efficient feather-degrading strains**. Feather-degrading bacteria were isolated from a feather waste dumping site in Shaoguan (Guangdong, China). Soil samples were serially diluted, inoculated in enriched medium, and incubated in a rotary shaker at 37 °C for 3 days. Bacterial suspensions were further plated onto feather powder plates and cultured at 37 °C for 5 days. Strains with strong growth were streaked onto feather meal plates, cultured at 37 °C, and single colonies picked. Isolates were then transferred to 1% CFM to test their feather degradation ability.

**Analysis of feather degradation rate**. Feather degradation rates were evaluated using the weight loss method[43]. After feathers were degraded by FDB, the fermented medium was filtered through Whatman No. 1 filter paper. Feather residue was thoroughly washed with double distilled water, dried at 65 °C, and then weighed to calculate weight loss. Results are expressed as percentage weight loss relative to the initial dry feather weight. All experiments were performed in triplicate.

**Determination of keratinase activity**. Keratinase activity was tested using 1% soluble keratin as substrate, according to a previously reported method, with some modification[44]. Soluble keratin (100 μL of 1% (w v$^{-1}$)) was added to 100 μL of diluted crude keratinase solution and incubated at 50 °C for 20 min. Hydrolyzation was stopped by the addition of 200 μL TCA and tubes centrifuged at 12,000 × g for 5 min. Aliquots of supernatant (100 μL) were pipetted into separate tubes containing 500 μL of 0.4 M Na$_2$CO$_3$ and 100 μL of Folin phenol reagent. The mixture was then incubated at 40 °C for 20 min and OD values measured at 660 nm. A tyrosine standard curve was constructed for quantification. One unit of keratinase activity was defined as the amount of enzyme needed to release 1 μg tyrosine from keratin per min.

**Determination of amino acid, protein, and sulfhydryl content**. Amino acid concentration was tested using ninhydrin reagent[45]. FDB culture broth was pre-cipitated using 20% TCA and 200 μL of supernatant mixed with 50 μL phosphate buffer (pH 8.04) and 50 μL 2% (w v$^{-1}$) ninhydrin reagent. The mixture was heated in water bath at 90 °C for 30 min, followed by addition of 950 μL distilled water. Absorbance was read at 570 nm to quantify the amino acids present in the hydrolysate according to a prepared isoleucine standard curve. Amino acid composition and content were further evaluated and quantified using an Amino acid analyzer A-300 advanced (MembraPure, Germany). A bicinchoninic acid (BCA) assay kit from TaKaRa (Shanghai, China) was used to determine the soluble protein concentrations in fermentation solutions, using bovine serum albumin as the standard. Each sample was assessed in triplicate.

The release of sulfhydryl groups into the FDB culture medium was determined spectrophotometrically, according to the method of Ellman[46]. DTNB (10 μL) and 0.1 M phosphate buffer (500 μL; 1 mM EDTA, pH 8.0) was added to 50 μL of extracellular broth mixture. Absorbance was measured at 420 nm and the concentration of sulfhydryl groups calculated.

**Detection of sulfite in feather culture medium**. Sulfite was detected by obser-vation of white precipitate formation on BaCl$_2$ addition and bubbles on HCl addition. The presence of sulfite was further confirmed by KMnO$_4$ decolorization[47].

**Applications in plant, animal, and microorganism culture**. Tilapia was used for animal culture experiments as follows. Healthy tilapia (weight: $6.9 \pm 0.2$ g) was randomly divided into three groups (three parallels in each group and 20 fish per parallel), fed with 45% feather meal (40% SCUT-3 cultured feather) feed, 45% full fish meal feed, and 22.5% fish meal plus 22.5% feather meal feed. Besides the 45% protein source, 53% flour, 1% CaHPO₄ and 1% vitamin and mineral premix were added into the feed mixtures. After eight weeks of culture, fish were weighed and the feed coefficients calculated for each group using the following formula:

$$\text{Feed coefficient} = \text{Feed consumption/weight gain} \times 100\%.$$

Rice and a succulent plant were used in the plant growth experiments with addition of 10% SCUT-3 cultured feather broth. Rice plants (nine per pot) and succulents (one per pot) were planted with the addition of 50 mg amino acids in 10% cultured feather broth in 50 g soil. Plants were watered every 3 days to keep the soil moist. The fresh weight of rice (g) was measured after 3 weeks.

*Bacillus subtilis* WB600-kerK was used in the microbial growth experiment with addition of 10% SCUT-3 cultured feather broth. *Bacillus subtilis* WB600-*kerK* was inoculated in 20 mL LB medium with addition of 3 mg amino acids in 10% cultured feather broth. After 24 h culture, the OD₆₀₀ value and the KerK enzyme activity were tested and record.

**SCUT-3 species identification**. The physical and chemical characteristics of SCUT-3 were evaluated according to the instructions in Bergey's Manual of Systemic Bacteriology. The ultra-scope images of 40% SCUT-3 cultured feather on different days were acquired using an ultra-high-resolution field emission scanning electron microscope (SEM) (Zeiss, German).

The SCUT-3 genome was sequenced by GENE DENOVO (Guangzhou, China). SCUT-3 16 S rRNA sequences were retrieved from the genome annotation results for constructions of the phylogenetic tree by distance matrix analysis using the neighbor-joining method with MEGA 7.0 software[48]. OrthoANIu was used to analyze the ANI of the SCUT-3 genome relative to the genomes of 1,222 other *Streptomyces* strains[49].

**Transcriptomes of SCUT-3 cultured in LB and feather medium**. SCUT-3 was cultured for 24 h in LB and 1% CFM in exponential growth phase. Total RNA was extracted using the RNAprep Pure Cell/Bacteria Kit, according to manufacturer's specifications (TIANGEN Biotech Co. LTD) and sequenced by LongseeMed (Guangzhou, China). Assembled transcriptome data were screened for differentially expressed genes using edgeR33 with a threshold of FDR < 0.05 and |log₂ FC | > 1, resulting in identification of 1,459 genes with significantly different expression levels. Gene Ontology (GO) and Kyoto Encyclopedia of Genes and Genomes (KEGG) pathway enrichment analysis of significant differentially expressed genes (DEGs) was conducted using clusterProfiler[50]. Selected DEGs ($n = 23$) were subjected to further verification by real-time qRT-PCR using the primers detailed in Supplementary Table 4, with 16 S rRNA levels used as an endogenous control to normalize gene expression levels. The $2^{-\Delta\Delta CT}$ method was used to estimate relative target gene expression levels, which are expressed as relative fold-change values. All samples were analyzed in triplicate. To determine the position of DEGs in the genome, transcription data were mapped to the genome sequence data. FGENESB was used for genomic operon prediction. DEGs and their associated operons were visualized in the genome using the IGV genome browser[51]. Based on the results of transcriptome analysis, nine protease genes (log₂ FC > 2) and one disulfide reduction related gene were detected at 0, 3, 6, 12, and 24 h by qRT-PCR.

**Overexpression of protease Sep39 in SCUT-3**. To further validate the key genes in feather degradation, an overexpression system was constructed in SCUT-3. The plasmid constructed in this study was introduced into *Streptomyces* sp. SCUT-3 according to the *Escherichia coli*–*Streptomyces* conjugation method reported previously[52]. The protease gene *sep39*, which has the highest up-regulation fold in transcriptome data, was used as the target gene. The overexpression plasmid pSET-*sep39* was obtained by inserting this gene into the *Nde*I/*Ecor*I sites of pSET152. Integrated plasmid pSET-*sep39* was transformed into the donor strain *E. coli* ET12567/PUZ8002, and then further transferred into SCUT-3 by conjugation. In order to detect whether the target gene was successfully overexpressed in SCUT-3, the relative expression of *sep39* was determined in the wild strain SCUT-3 and the overexpression strain SCUT-3-*sep39* by qRT-PCR. Keratinase activity, degradation rate, and soluble protein and amino acid content were also determined in SCUT-3 and SCUT-3-*sep39* by the method mentioned above.

**Analysis of SCUT-3 reducing power**. To test the reducing power of SCUT-3, SCUT-3 was inoculated in 50 mL 1% CFM for 24 h and centrifuged to collect the cell-free supernatant (extracellular component) and cell pellets (living cells). Cells were also disrupted by sonication and centrifuged to collect intracellular components in the supernatant. Extracellular supernatant (10 mL), intracellular component, and living cells ($10^9$ CFU) in 10 mL PBS were added into 50 mL tubes containing 2 mM GSSG as a disulfide bond substrate and incubated at 37 °C. Samples were collected at different time points and released sulfhydryl groups tested using the method described above.

**Statistics and reproducibility**. Mean from three independent biological experiments was presented in each plot, in which error bars represented standard deviation. A number $n$ suggested biological replications indicated in the figure legends. Statistical significance was assessed in GraphPad Prism 8.0. Unpaired two-tailed Student's t test was used for comparison of two experimental groups. A $P$-value of 0.05 was deemed statistically significant and statistical details are found in the figures and figure legends.

**Reporting summary**. Further information on research design is available in the Nature Research Reporting Summary linked to this article.

## Data availability

The 16s rRNA and genomic sequences of *Streptomyces* sp. SCUT-3 were submitted to the NCBI accession numbers MK743936.1 and CP046907, respectively. The RNA-seq data used in this study are deposited in the National Center for Biotechnology Information SRA database (SRA accession no. PRJNA611875). The authors declare that all data supporting the findings of this study are available within the article, its Supplementary Information file and from the corresponding author on reasonable request. The source data underlying the graphs and charts presented in the main figures are shown as Supplementary Data 1–5.

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

## Acknowledgements

This work was supported by Dedicated Fund for Promoting High-Quality Economic Development in Guangdong Province (Marine Economic Development Project) to Dr. Xiao-Chun Luo (No. GDME-2018c008) and Dr. Yong-Hua Wang (No. GDOE [2019] A20), and Guangdong Natural Science Foundation to Dr. Ye Ke (No.2019A1515011089), and Special Fund for Zhanjiang Science and Technology Development to Dr. Jia-Zhou Li (2018A01011).

## Author contributions

Z.-W.L. and X.-C.L. designed the experiments; Z.-W.L. conducted most experiments; S.L. performed experiments on gene overexpression; Z.-W.L., Y.K., J.-J.D., M.-S.Z., D.-L.L., J.-Z. L. and X.-C.L. analyzed the data; Z.-W.L. and X.-C.L. wrote the manuscript. All authors read and approved the final manuscript.

## Competing interests

The authors declare no competing interests.
