## [Peer Review File · Communications Biology]

Reviewers' comments:

Reviewer #1 (Remarks to the Author):

In the present study, a lot of work has been done by the authors and they have isolated a very good strain that can degrade even up to 40% of feather waste. Such a strain will find a number of industrial applications as low levels of feather degradation is a major bottleneck in industrial level feather waste hydrolysis. The given strain degrades feather into products containing high amounts of amino acids and peptides that were demonstrated to be a good nitrogen source in rice plant, fishes and in *B.subtilis*. SEM was also performed to study the feather colonization by given strain. However, this application of feather meal is long known and a lot of literature is already available. The need of hour is that someone proposes time bound sequence of events during feather degradation which will include studies such as transcriptomics, secretome analysis, metabolomics, deletion and complementation studies.

In the present study, authors have attempted to elucidate the mechanism of feather degradation in this strain. However, only transcriptome analysis has been performed that too at a single time point. They have reported 19 extracellular protease genes that are significantly up-regulated in feather medium which are further confirmed by qRT-PCR. Transcriptome analysis needs to be done even at initial time intervals of colonization. But this transcriptome analysis may not reflect the same at translational level and validation of their role in feather degradation is essential to elucidate mechanism as up-regulation of certain gene may not be directly related to the process. For this, deletion mutant analysis followed by complementation studies need to be performed which is elaborate and time consuming. At least secretome or metabolome studies should be done to obtain some reliable results in elucidating the mechanism of feather degradation in this strain. Additionally some experiments are done such feather degradation by recombinant keratinase in presence of DTT which is not relevant to the present study. Looking at the observations following major revisions are suggested where experimentation is required:

1. Transcriptome analysis should be done at different time points within 24h. If many points cannot be taken it should be performed at least at early hours of colonization.
2. Transcriptome data need to be complemented with at least secretome or metabolome data if not deletion and complementation studies to pin point key enzyme active.
3. The studies done with recombinant keratinase in presence of reductants is irrelevant to the present manuscript because such mechanisms have already been cited in literature several time. So it should be removed as it is changing the focus of the script.

Reviewer #2 (Remarks to the Author):

With the development of chicken breeding industry, more and more feather wastes are produced and

urge for the development of new recycling technology. FDBs are most promising organisms for feather utilization, while limited by low efficiency and food and feed safety. This article provides a new soil isolated bacteria which can efficiently degrade feather and its fermented products is safe for animal feed proved by fish feed addition experiments. I think this is an important finding for feather waste recycling industry. By comparative transcriptomes and other verification experiments, the authors provide new knowledges for FDB's feather degrading mechanisms. To my opinion, this is the first article which systematically describes a FDB's feather degrading process, including its colonization biology, disulfide bond reduction, keratin hydrolysis, peptides and amino acids transportation, and so on. I agree with the authors' suggestion that the future research ought to pay more attention on the disulfide bond reduction mechanisms besides keratinases. We hope to see the authors' further works on this bacteria's genetic modification and more details of its feather degrading mechanisms. I believe this article will give reference for other researchers in feather recycling field and suggest its publication in Communications Biology.

Reviewer #3 (Remarks to the Author):

In this paper Li et al. describe a bacterium capable of degrading feather. Authors identify this bacterium as one of the most active among a set of other bacteria using a screen tests this this complex material. Later, authors determined the degradation rate, and after confirming the rate was high, authors proceed with the analysis of the bacterium, that include the 16S rRNA analysis, genome and transcriptome sequencing, morphology analysis, etc. Finally, authors suggest a number of genes found to be expressed most like implicated directly or indirectly in feather degradation.

Although the degradation of feather is recognized as complex, the degradation mechanisms are known, with the participation of proteolysis (including proteases/peptidases and keratinases), sulfitolysis, deamination, etc. Given that those mechanisms are also found in the bacterium herein reported, the outcomes are not novel and the novelty limited.

Below a number of major comments are given.

1. I did not find citation of the accession number for the genome sequence.
2. Although the analysis of the bacterium is appreciated, I recommend performing a proper taxonomic analysis and deposition of the bacterium in a culture collection.
3. In Figure 6 authors provide a schematic diagram of feather utilization mechanism. However, this mechanism is very general and known. Since authors sequence the genome and transcriptome I recommend authors providing in silico and experimental data about the specific genes and enzymes directly participating in the feather degradation. This will provide novel insights into the feather degradation mechanisms which may open new opportunities to other researchers interested in improving feather degradation.
4. Authors used the feather degradation products as source for recombinant protein production. For the tests authors used a culture of recombinant *Bacillus subtilis* expressing a keratinase. Increase expression of keratinase is expected when using a feather-based material, but for other enzymes not related to feather degradation the utilization of fermented feather broth is unclear. Author should use different expression systems with different heterologous proteins.
5. Authors mentioned in page 5 that sulfite at concentrations as high as 8 M improved feather degradation, but did not help degradation below 80 mM. Later authors found that the bacterium under

investigation produced sulfite intracellularly at concentrations below 0.05 mM (see Figure 3b). According to the *in vitro* experiments, one would hypothesize that at such low concentration of sulfite *in vivo* would not favor feather degradation to support the high degradation rates found by authors. An explanation is needed.

6. A number of proteases and peptidases were found to be expressed in cultures with feather. I suggest authors providing biochemical evidences about which of these enzymes play a major role in the initial degradation of feather.

7. Authors mentioned that iron and aeration are essential for improving feather degradation. Although such conditions are in general favoring microbial metabolism, the direct implication of iron and O₂ in key enzymes directly involved in feather degradation is missing. Mentioning that iron may provide ferric iron for the Fe-S clusters is known and this assumption is not giving any novel information.

8. Authors also analysed the genome and transcriptome data for genes implicated in general metabolism, spore formation, cell division, etc. While of interest, a direct link with feather degradation is missing, and without this information this section does not provided any novel information. In lanes 284-286 authors mentioned that two esterases, among other enzymes, may be implicated in lipid metabolism; this should be biochemically proved as esterases compared to lipases are not well performing towards lipids.

9. A proper statistical analysis is missing, as many of the data are given in average values without standard deviation values.

10. References need extensive revision; for example, in some cases journals names are completed and in other cases abbreviated, microorganism names in titles are not in many cases in italics, etc.

Reviewer #1:

1. Transcriptome analysis should be done at different time points within 24h. If many points cannot be taken it should be performed at least at early hours of colonization.

Answer: Thanks for this suggestion. As we discussed with Dr. Akhtar last time, we believe more omics would not much help the reveal of feather degradation mechanisms without further genetical modification verification. We would like to provide different time points qPCR of some important genes. In the revised paper, we have added different time points (0, 3, 6, 12 and 24 h) qPCR of nine extracellular proteinase genes and one disulfide bond reduction related gene *cdo1* as we showed in Fig 4c. These results were described in a separated paragraph. Please see line 228-237.

2. Transcriptome data need to be complemented with at least secretome or metabolome data if not deletion and complementation studies to pin point key enzyme active.

Answer: As we discussed with Dr. Akhtar, genetic manipulation of *Streptomyces* strains, especially a new species of *Streptomyces* genus, is much more difficult than other model microorganisms. We have recently established a reliable overexpression system for *Streptomyces* sp. SCUT-3. With this system, we have successfully over-expressed a hypothesis key protease Sep39 and found its overexpression could improve the keratinase activity and feather degradation efficiency of SCUT-3, which confirmed this enzyme is an important protease for keratin hydrolysis. We have described these results in a separated paragraph of the revised article, please see line 238-255. More key genes' function in the feather degradation process is under verification in our lab and will be published in our further article.

3. The studies done with recombinant keratinase in presence of reductants is irrelevant to the present manuscript because such mechanisms have already been cited in literature several time. So it should be removed as it is changing the focus of the script.

Answer: Thanks for the suggestion, we have removed the corresponding content in the revised manuscript.

Reviewer #2:

We thank reviewer #2's positive affirmation of our work.

Reviewer #3:

1. I did not find citation of the accession number for the genome sequence.

Answer: The genome data of *Streptomyces* sp. SCUT-3 has been submitted to NCBI, and the accession number is CP046907. We have added this number in the revised manuscript. Please see line 129.

2. Although the analysis of the bacterium is appreciated, I recommend performing a proper taxonomic analysis and deposition of the bacterium in a culture collection.

Answer: The physiological and biochemical character and ANI analysis showed SCUT-3 is a new species of *Streptomyces* genus. The systematic evolution tree was constructed according to 16s rDNA sequences and showed it was genetically closest to *Streptomyces thermolineatus*, please see supplementary Fig.1. We have deposited this strain in the Guangdong Microbial Culture Collection Center (GDMCC), and the strain number is GDMCC No: 60612. This information was added in the revised paper, please see lines 143-144.

3. In Figure 6 authors provide a schematic diagram of feather utilization mechanism. However, this mechanism is very general and known. Since authors sequence the genome and transcriptome I recommend authors providing in silico and experimental data about the specific genes and enzymes directly participating in the feather degradation. This will provide novel insights into the feather degradation mechanisms which may open new opportunities to other researchers interested in improving feather degradation.

Answer: The mechanism of feather degradation has dispersedly reported in many articles. The schematic diagram in this paper systematically described the feather degradation process of SCUT-3, including the possible key genes involved, such as reductants production and transportation, proteases and amino acids/peptides transporters, iron uptake regulon and cell respiration, and so on. Some of these genes and processes are firstly confirmed in our work. We agree with the reviewer's suggestion to provide further experimental data of specific genes. As we response to reviewer #1's Q2 above, we have provided the overexpression data of a key protease Sep39 in the revised manuscript, please see line 238-255 and 523-534.

4. Authors used the feather degradation products as source for recombinant protein production. For the tests authors used a culture of recombinant *Bacillus subtilis* expressing a keratinase. Increase expression of keratinase is expected when using a feather-based material, but for other enzymes not related to feather degradation the utilization of fermented feather broth is unclear. Author should use different expression systems with different heterologous proteins.

Answer: The promoter of the recombinant keratinase in *Bacillus subtilis* is a

constitutive promoter and is not induced by feather. We just used this recombinant keratinase as an example of the recombinant protein production in the culture medium with addition of fermented feather broth, which may provide more amino acids and peptides ingredient for protein synthesis. Actually, the phenomenon of more recombinant protein production after the fermented feather broth addition in the culture medium is observed not only in *Bacillus subtilis*, but also in recombinant *P. pastrois* and *E. coli* in our lab. We will publish these data in another article.

5. Authors mentioned in page 5 that sulfite at concentrations as high as 8 M improved feather degradation, but did not help degradation below 80 mM. Later authors found that the bacterium under investigation produced sulfite intracellularly at concentrations below 0.05 mM (see Figure 3b). According to the in vitro experiments, one would hypothesize that at such low concentration of sulfite in vivo would not favor feather degradation to support the high degradation rates found by authors. An explanation is needed.

Answer: Figure 3b is not sulfite production figure, it showed that the living cells of SCUT-3 could release sulfhydryl content by reducing the added oxidized glutathione, not sulfite. The sulfite production by SCUT-3 is indeed lower than 80 mM in the feather culture medium and we hypothesize the compact contact of SCUT-3 mycelia and feather could form locally high sulfite concentration according to the ultra-microstructure observation. Please see this explanation in line 189-192. This hypothesis that sulfite production is important for feather's disulfide bond reduction has been further confirmed in our lab. The knock-down of a key enzyme Cdo1 for sulfite production significantly reduces disulfide bond reduction and feather degradation efficiency of SCUT-3. More details of disulfide bond reduction

mechanism of SCUT-3 will be published in our further article.

6. A number of proteases and peptidases were found to be expressed in cultures with feather. I suggest authors providing biochemical evidences about which of these enzymes play a major role in the initial degradation of feather.

Answer: We are now over-expressing 19 secretory proteases in SCUT-3 to see which proteases play the major role of feather keratin hydrolysis. In the revised article, we added the overexpression data of Sep39, one of these 19 proteases, and confirmed its function in feather degradation. Please see line 238-255 and 523-534.

7. Authors mentioned that iron and aeration are essential for improving feather degradation. Although such conditions are in general favoring microbial metabolism, the direct implication of iron and O₂ in key enzymes directly involved in feather degradation is missing. Mentioning that iron may provide ferric iron for the Fe-S clusters is known and this assumption is not giving any novel information.

Answer: We found the Fur regulon for iron uptake is significantly up-regulated in feather medium cultured SCUT-3. We agree that this phenomenon is not theoretical novel information for microbial. While for feather degradation bacteria, this is firstly reported that iron addition could improve bacteria's feather degradation. The iron and aeration are two important factors for SCUT-3's industrial application for feather degradation. According to these finding, we have designed a specific aeration system in our pilot feather fermentation facility and we found mixing the feather with rice husk could improve the oxygen supply and feather degradation efficiency. We believe these finding will give reference for other FDB's development and research.

8. Authors also analysed the genome and transcriptome data for genes implicated in general metabolism, spore formation, cell division, etc. While of interest, a direct link with feather degradation is missing, and without this information this section does not provided any novel information. In lanes 284-286 authors mentioned that two esterases, among other enzymes, may be implicated in lipid metabolism; this should be biochemically proved as esterases compared to lipases are not well performing towards lipids.

Answer: We believed this information is useful novel information. With this information, we found the addition of starch/glucose or glycerol can directly inhibit the feather degradation through metabolism regulation mechanism. After the feather degradation pathway is firstly evoked, the addition of starch at day 2 will not inhibit feather degradation, while improve the amino acids and peptides yield by reducing bacteria's amino acids catabolism and using glucose as its carbon source. The key transcription factor involved in SCUT-3's metabolism regulation is under investigated in our lab and manipulation of its metabolism will improve the feather's utilization efficiency.

9. A proper statistical analysis is missing, as many of the data are given in average values without standard deviation values.

Answer: Statistical analyses have been performed and the methods are added in line 544-546. The standard deviations are added in the corresponding figures.

10. References need extensive revision; for example, in some cases journals names are completed and in other cases abbreviated, microorganism names in titles are not in many cases in italics, etc.

Answer: We have carefully edited the references and all microorganism names.

We have carefully examined the manuscript and corrected other mistakes and typographical errors. We look forward to seeing the acceptance of the revised manuscript by *Communications Biology*.

Reviewers' comments:

Reviewer #1 (Remarks to the Author):

Authors have done extensive revision taking into account all the queries. The manuscript is satisfactorily revised.

Reviewer #3 (Remarks to the Author):

This is a revised version of the paper which I reviewed. I appreciate the effort done by the authors and the additional data provided. Still I have a number of concerns that are below:

Authors mentioned in the abstract that "... this study presents a green method for recycling feather waste". Although I agree with the authors that the strain here described degrades feather, no optimized method allowing optimal feather waste is presented. Therefore, I recommend revising this phrase.

In page 3, authors provided information about degradation rates. Statistical significance or standard deviations are needed and should be included.

Lane 158: "... in the presence or absence of feather" instead of "... in the presence of absence of feather".

Pages 5 and 6. Authors commented about the sulfite production issue that I raised in previous report. Still it is not clear how this bacterium reached so high degradation capacity when they claim that only after adding 8 M of Na₂SO₃ keratinase can improve feather degradation, which was not observed below 80 mM. The reasoning that a local enrichment of sulfite can occur at the mycelia is not supported by the presented data. A clear reasoning for the very high degradation capacity is needed in this respect, as it is expected that 8M sulfite cannot be locally produced in mycelia.

Page 6. Authors discussed that MSH may be a cysteinyl group involved in feather sulfitolysis and that GSI/BCA may be responsible for free cysteinyl group transportation. However, this is based on examining the expression level of few genes. This is why authors claim this needs to be verified in a further study. Therefore, the exact implication and role of MSH in feather degradation is needed; also which of the MSH or sulfite plays a major role in degradation need to be clarified.

The effort in relation to expression of key proteases is appreciated. Authors presented data demonstrating that Sep39 overexpression improved degradation. However, the exact role of this protease compared to others is not mentioned. Authors only discussed in lane 227 that is involved in keratin hydrolysis, but I assume this is the case of many other proteases.

Did authors check the expression of genes relevant for acidification? This may be of interest as keratin hydrolysis may be improved under acidic conditions.

Although the analysis of expression level of genes involved in iron uptake, aeration, spore formation, cell division and metabolism is appreciated, still specific determinants of high degradation efficiencies need to be established as many of the genes found to be expressed are also expressed by other microorganisms in conditions different to those examined here.

Reviewer #1:

We thank reviewer #1's positive affirmation of our work.

Reviewer #3:

Authors mentioned in the abstract that "... this study presents a green method for recycling feather waste". Although I agree with the authors that the strain here described degrade feather, no optimize method allowing optimal feather waste is presented. Therefore, I recommend revising this phrase.

Answer: We have modified this sentence in the revised manuscript according to this suggestion. Please see line 21-23.

In page 3, authors provided information about degradation rates. Statistical significance or standard deviations are needed and should be included.

Answer: Standard deviation Bars had already added in the Fig. 1a in the last revision manuscript. In this revised manuscript, the standard deviations of the degradation rate have been added, please see line 72, 84, 85 and 387.

Lane 158: "... in the presence or absence of feather" instead of "... in the presence of absence of feather".

Answer: Thanks to the reviewer for his suggestion. We have modified the phrase in the revised manuscript, please see line 155-156.

Pages 5 and 6. Authors commented about the sulfite production issue that I raised in previous report. Still it is not clear how this bacterium reached so high degradation capacity when they claim that only after adding 8 M of Na₂SO₃ keratinase can

improve feather degradation, which was not observed below 80 mM. The reasoning that a local enrichment of sulfite can occur at the mycelia is not supported by the presented data. A clear reasoning for the very high degradation capacity is needed in this respect, as it is expected that 8M sulfite cannot be locally produced in mycelia.

Answer: We think our description of Na_2SO_3 addition experiment results may cause reader's confusion. Actually, this experiment was not carried out in vivo, but in vitro with a recombinant keratinase KerK. We have done different dilution of Na_2SO_3 and the results is showed as following:

As the figure above, the addition of Na_2SO_3 with the concentration above 80 mM could significantly improve KerK's in vitro degradation of feather. With these data, we only try to let the reader know Na_2SO_3 can really help feather's disulfide bond's breakdown. While the concentration of 8 M may mislead reader to think that the concentration needs to be so high. Actually, 80 mM is enough for a significant improvement though the amplitude is not as high as 8 M. We did try to titrate the concentration of Na_2SO_3 in feather fermentation supernatant, but failed. The titration method is not sensitive for low concentration, but our qualitative experiment by BaCl_2 precipitation, HCl bubble and KMnO_4 bleaching did show the existence of Na_2SO_3 in SCUT-3 fermented feather. We provide a possible explanation that comparatively high sulfite concentrations could be produced locally in the compact contact surfaces

between SCUT-3 mycelia and feather barbules. Our further experiments (data not shown in this paper), which showed the overexpression of *cdol* could significantly improve the bacteria's reducing power and knockdown of *cdol* significantly reduced its feather degradation efficiency, also confirm the sulfite production is important for SCUT-3's disulfide bond breakdown.

To eliminate the possible confusion, we replace the 8 M Na₂SO₃ data with 80 mM data in figure 3c and the degradation rates of different concentration Na₂SO₃ addition are added in supplementary figure 6. We hope this revision may help reader understand what we really want to express. Corresponding description of this revision was made, please see line 183-186.

Page 6. Authors discussed that MSH may be a cysteinyl group involved in feather sulfitolysis and that *gsiDBCA* may be responsible for free cysteinyl group transportation. However, this is based on examining the expression level of few genes. This is why authors claim this needs to be verified in a further study. Therefore, the exact implication and role of MSH in feather degradation is needed; also which of the MSH or sulfite play a major role in degradation need to be clarified.

Answer: Yes, all these *gsiDBCA* and MSH are speculations. We only confirm, besides sulfite, cysteinyl group also attends feather disulfide bonds breakdown. The DTNB addition significantly inhibits SCUT-3's feather degradation, but we don't know how the free cysteinyl groups is produced and out-transported during its degradation process. Other groups had reported previously *Streptomyces* genus does not have GSH system but substituted by MSH. MSH synthesis and *gsiDBCA*' transportation is under testing in our lab and results is not got yet. We hope our further work will reveal the corresponding mechanism.

The effort in relation to expression of key proteases is appreciated. Authors presented data demonstrating that Sep39 overexpression improved degradation. However, the exact role of this protease compared to others is not mentioned. Authors only discussed in lane 227 that is involved in keratin hydrolysis, but I assume this is the case of many other proteases.

Answer: Thanks for the confirm of our effort. Yes, of course, we agree the reviewer that keratin hydrolysis by SCUT-3 involves different many proteases' co-operation, not only by one key protease. Its hard to say which one is most important, while others are not so important, so we did not discuss too much their exact role of each different extracellular proteases. Indeed, we did also overexpress other extracellular protease genes to see which one could most efficiently improve SCUT-3's feather degradation and figure out their amino-acid peptide bond substrate preference to see how they complement with others to achieve a higher hydrolysis efficiency. We will publish these works in subsequent articles.

Did authors checked the expression of genes relevant for acidification? This may be of interest as keratin hydrolysis may be improved under acidic conditions.

Answer: Yes, previous reports showed acidification may exist in some fungi's feather degradation. For this streptomycetes, it likes and works better under alkaline circumstance than acidic one, and we did not find the phenomena of acidification.

Although the analysis of expression level of genes involved in iron uptake, aeration, spore formation, cell division and metabolism is appreciated, still specific determinants of high degradation efficiencies need to be established as many of the

genes found to be expressed are also expressed by other microorganisms in conditions different to those examined here.

Answer: Yes, we agree with this comment. We only provide some possibility of potential processes involved in SCUT-3's feather degradation and a lot of works need to carry out for further confirmation. These genes may or may not only involved in feather degradation and may have different patterns in other FDBs under different conditions. We just present them here and hope they may give some reference for other FDBs research.

We have carefully examined the manuscript and corrected other mistakes and typos. We look forward to seeing the acceptance of the revised manuscript by *Communications Biology*.

REVIEWERS' COMMENTS:

Reviewer #3 (Remarks to the Author):

This paper is a revised version of a previously submitted manuscript, which I revised. Authors consider most of my previous comments. Here are some further minor comments:

Please remove the word "ignorance" in the abstract; possibly better "limited knowledge" or similar.

Authors discussed about the Sep39 protease. They should classified the protease according to classification by MEROPS database and Pfam peptidase domains, so readers can clearly see the protease/peptidase clan to which is associated.

Reviewer #3 (Remarks to the Author):

Please remove the word “ignorance” in the abstract; possibly better “limited knowledge” or similar.

We have revised the word "ignorance" to "limited knowledge" in the revised manuscript. Please see line 13.

Authors discussed about the Sep39 protease. They should classified the protease according to classification by MEROPS database and Pfam peptidase domains, so readers can clearly see the protease/peptidase clan to which is associated.

We have supplemented the classifications of all mentioned proteases according to the MEROPS database and Pfam peptidase domains in the revised Supplementary Information (Please see Supplementary Table 4). This information was also mentioned in line 218-220.